# A sustainable working life in the car manufacturing industry: The role of psychosocial factors, gender and occupation

Kristina Gyllensten *, Kjell Torén, Mats Hagberg, Mia Söderberg

Department of Occupational and Environmental Medicine, Sahlgrenska Academy and University of Gothenburg, Gothenburg, Sweden

☯ These authors contributed equally to this work.
* kristina.gyllensten@amm.gu.se

**Data Availability Statement:** Data cannot be shared publicly because information about health and symptoms are regarded as sensitive information, and when sharing such data there has

## Abstract

### Aims

In order to add to the existing knowledge about factors associated with retirement timing, in the car industry, it is useful to consider the psychosocial working conditions prior to retirement. This case-control study aimed to investigate relationships between psychosocial job factors and extended work after the age of 62 years among workers in the car industry in Sweden.

### Methods

A study invitation with a survey was sent to workers in one of Sweden's largest car manufacturing company, who were employed 2005–2015 and either retired at the age 55–62 years or working at 63 years or older. Psychosocial variables such as job demand-control (JDC) and effort-reward imbalance (ERI) were recorded through the survey. Multiple logistic regression models were used to investigate associations between psychosocial variables and retirement in 572 cases that had continued to work $\geq$ 63 years, and 771 controls who had retired at 62 or earlier.

### Results

No associations were found between JDC-variables and retirement in the total sample or gender stratified analyses, but high demands-low control (high strain) was related to retirement before the age of 63 years in blue-collar workers. In contrast, high strain was related to continuing to work after 62 years for white-collar men and, high ERI was associated with extended work for the total sample of white-collar workers, and white-collar men, however these effects became non-significant in fully adjusted models.

### Conclusions

The relationships between psychosocial factors and extended work after 62 years were inconsistent, with high strain being related to retiring earlier for blue-collar workers.

to be an approval from a Swedish Ethical committee (according to Swedish law). However, anonymised data is available with an approval from an ethical review board. For data requests, contact: Department of occupational and environmental medicine, Gothenburg University, amm@amm.gu.se or Kristina Gyllensten, Department of occupational and environmental medicine, Gothenburg University, kristina.gyllensten@amm.gu.se. The name of the data set is 'The Volvo Work Ability Study.'

**Funding:** The study was funded by AgeCap – a centre for aging and health at Gothenburg University and by the Swedish Research Council for Health, Working life and Wellfare. The funders did not have any involvement in the study, the writing of the manuscript or the decision to submit the paper for publication. There was no additional external funding received for this study.

**Competing interests:** Volvo Cars provided contact details, age, details of level of employment and time of employment, of all employees that had worked/ and or retired at Volvo between the years of 2005-2015 that were 55 years or older. Volvo did not have any further involvement in conducting the study or in the decision to publish the study. This does not alter our adherence to PLOS ONE policies on sharing data and materials.

## Introduction

The ageing European population calls for the implementation of policies that promote healthy ageing and a sustainable working life [1]. Indeed, governments have implemented policies aiming to encourage later retirement, including raising the age for state pensions and legislation against age and disability discrimination [2]. However, there has been some concern regarding polices to extend working life and closure of early exit pathways. Certain groups within the population, with very demanding jobs, may still need the option of early retirement [3]. Continuing to work at an older age is a complex issue, and there is limited knowledge of push and pull determinants. A few identified 'push factors' include chronic diseases, physical demands and poor working conditions and 'pull factors' constitutes one's spouse not working, care-taking of relatives and leisure time expectations [4,5]. Norms about working and retiring, economic incentives, attitudes at the workplace, work satisfaction, social relationships at work and home are also important factors for an extended working life [6].

The car manufacturing industry is one of the largest industries worldwide, and is viewed as one of the most important and strategic industries in the manufacturing sector. Although a substantial amount of research have been conducted within the car industry [7] there is a lack of knowledge of factors influencing the timing of retirement in this sector. The car industry provides a large variety of jobs, both manual jobs at the manufacturing sites and office work involving administration, design development and marketing. The role of workplace factors in timing of retirement most likely differs between blue- and white-collar workers, since performing different jobs and, thus, face different working conditions [8]. There is evidence that white-collar workers with higher education tend to work to a later age compared to blue-collar workers [9]. A Swedish study of retirement intentions found that white-collar workers, in high demand -high control jobs, would consider continue to work after the statutory age for old age retirement, whereas most blue-collar workers wanted to retire early [10]. Another study in workers aged 45+ [11] found that job resources (job control and social support) were positively related to work enjoyment and negatively related to early retirement intention. These relationships were found in both blue- and white-collar workers, but were stronger in blue-collar workers. There were also associations between job demands and early retirement intention. It is further plausible that gender plays a role regarding factors determining retirement timing. A previous study found that the strength of a number of push and pull factors relating to retirement decisions were different for men and women. Demanding jobs was a push factor for women and socially rewarding jobs was a pull factor for men [12]. In Sweden, women retire on average one year earlier than men ($\bar{x}_{women}$ = 63 years; $\bar{x}_{men}$ = 64 years) [13]. Women tend to have lower level positions and the working conditions for men and women may differ even in the same profession and organisation. In addition, women spend more time taking care of home and family compared to men [14].

There is extensive evidence that psychosocial factors at work influence physical and mental health [15,16]. Current knowledge about stressing psychosocial work environment mainly relies on two theoretical models, the demand–control–support model (JDC-S model) [17], and the effort–reward imbalance (ERI) model [18]. Indeed, psychosocial factors have been related to retirement timing, and several studies have found associations between low levels of job control and early retirement [19]. Similarly, job strain (combination high demands-low control) have been associated with intentions to leave work early [20]. Effort-reward imbalance has also been associated with intentions to early retirement [21]. A longitudinal study conducted in an English population aged 50+, found that high job demands were associated with preferences for early retirement, and high decision authority was associated with preferences for later retirement [5]. Another study [22] illustrated that stressful jobs decreased the

risk of early retirement. One proposed explanation was that stressful jobs were seen as more meaningful, stimulating and satisfying. However, it was also found that men with low levels of job control retired earlier compared to men with high levels of control.

### Aims

In order to add to the existing knowledge about factors associated with retirement timing, in the car industry, it is useful to consider the psychosocial working conditions prior to retirement.

This case-control study therefore aims to investigate relationships between psychosocial job factors, using the JDC- and ERI-models, and extended work after the age of 62 years among workers in the car industry in Sweden. The age of 62 years as criteria was selected because in this particular company, due to beneficial retirement schemes, there is no particular economic gain by working after 62 years of age. Consequently, early retirement could indicate that other factors, such as the work environment, could be associated with the retirement decision.

## Methods

### Study population

The present study is a case-control study, conducted as part of "AgeCap—a center for aging and health". Potential study subjects constituted all individuals employed at Volvo Cars at sites located in the West county of Sweden during 2005–2015 and either retired at the age 55–62 or working ≥63 years during the observation years. Access to personnel who fulfilled inclusion criteria was available to us through Volvo's staff registers. In total, 3025 persons fulfilling inclusion criteria and with a valid postal address, were invited to the study. Out of those, 1871 men and women returned filled-in questionnaires, yielding a response rate of 61.9%. Some subjects completely missed fill-in psychosocial questions, and were thus, deleted. As job demand-control and effort-reward imbalance and some of the covariates, were analysed using sum scores, subjects with <50% missing items received imputed values, described under statistical analysis. Subjects lacking >50% filled-in items per each variable were excluded. Since the amount of subjects with >50% missing items varied between JDC and ERI, calculations for these psychosocial dimension were carried out in separate subgroups. The final sample used for JDC analyses constituted of 1320 subjects and in the ERI analyses 1305 subjects. This study has been approved by the Regional Ethical Review, Gothenburg, Sweden (dnr 371–15). The consent was written.

Through the register we also obtained information on employment status, i.e. if being currently active at work or retired, year of retirement and year or birth from which age at retirement could be calculated. This information was used to categorize staff into cases or controls. The cases were employees who during the observation period had either retired ≥63 years, or who were currently working and aged ≥63 years. Controls constituted all study base subjects who retired at 55–62 years. We defined cases as subjects continuing to work ≥63 years, rather than those retiring early, in order to emphasize work dimensions that benefit continued work. Individuals who had been laid off were not included in the study.

### Questionnaire

All potential participants were sent a questionnaire by post. The questionnaires recorded demographics, occupational history and position, shift work, physically and psychosocial work conditions, retirement circumstances, previous diseases, stress at home and life events. The same variables were measured for cases and controls, but the wording slightly differed since

cases could be either retired or still working, while all controls were retired. The cases were asked to consider their current working environment, or the working environment at their last workplace at Volvo. The controls were asked to consider the working environment at their last workplace at Volvo.

## Psychosocial variables

Job demand-control was measured with the Swedish Demand-Control-Support Questionnaire (DCSQ) [23]. For the purposes of analysis, all demand and control items were both positively inverted, i.e. a high scores equated either high demands or high control, and then tallied separately. Median scores for demand and control were 12 and 20, respectively. Both variables were dichotomized into high or low by the median values of the distributions, and combined into: *high strain* (high demand-low control), *active* (high demand-high control), *passive* (low demand-low control) and *low strain* (low demand-high control). In the regression analyses low strain was used as reference [24].

Reward was assessed using the Effort-Reward Imbalance at Questionnaire—short version (ERI-S) [25,26]. Effort and reward items were positively inverted and then summed up. According to standard procedure, a ratio value was created ($\Sigma_{effort}/(\Sigma_{reward^*0.4286)}$) and then dichotomized by the established cut-off 1.0, where a score above the cut-off represent reciprocity between work efforts and received rewards. ERI-ratio scores <1.0 were used as reference value.

## Statistical analysis

The statistical software package SAS version 9.2 for Windows (SAS Institute; Cary; NC) was used for all regression analyses. Multiple logistic regression models were used to investigate associations between psychosocial variables and extended work $\geq$ 63 years of age. Several available covariates could be important in the analyses of associations between psychosocial work conditions and retirement, but since our cohort size is fairly small and some analyses are stratified by gender or blue-/white-collar workers, we wish to slim the models by only including covariates relevant for our analyses. We therefore conducted, stepwise purposeful selection as proposed by Hosmer & Lemeshow [27] for associations between our psychosocial variables of interest and retirement at 62 years of age (yes/no). Correlation coefficient values > 0.4 was considered as co-linearity and cut-off for inclusion was Likelihood ratio 0.25. The potential covariates considered were: civil status, age of partner in relation to the participant, country of birth, retirement during a year of recession, being offered a beneficial retirement deal, if retirement was the persons own decision, amount of years since retirement, previous diseases, work ability at the time of retirement, need for recovery after work, life events, stress at home, frequency of heavy physical demands, quality of leadership, leadership position and social support at work.

According to established procedure the selection process began with checking for co-linearity. Retirement during economic recession, being offered a beneficial pension deal and amount of years retired, displayed co-linearity. Since retirement during recession year displayed strongest effect in relation to retirement, this variable was chosen for the next step. We then carried out univariate analyses of each potential covariate and retirement as outcome. All variables meeting the cut-off in the univariate analyses constituted the full model. These variables and main psychosocial measures were then entered together in a logistic regression analyses. Variables with Likelihood ratio >0.25 were excluded. The excluded variables were then added back one at a time and reinserted to the model if meeting inclusion criteria. The remaining variables constituted the reduced model. Since main effects in the reduced models were only reduced with 4.4%, i.e. less than criteria of 15%, compared to the full model, the reduced model was kept.

After concluding the selection process the following confounders remained: age difference to partner, frequency of heavy physical demands, leadership quality, having a leadership position, retired during an economic recession, whether it was the person's own decision to retire and life events.

Age difference to partner was captured by one item in the survey about spouse/partners age with was compared to the study participants age, and analysed as a categorical variable: younger, the same age or older. Frequency of heavy physical job demands was measured with an instrument developed by the Swedish work Environment Authority and constituted seven items inquiring about physical work tasks and a response scale (1–6): 1 = "No not at all" to 6 = "Almost all the time", sample item: "Does your job require that you a certain amount of the time work purely physically?". Leadership quality was measured with standard items from the Copenhagen Psychosocial Questionnaire (COPSOQ) defining "leader" as the participant's immediate superior [28]. Having a leadership position (yes/no) was meaured by one item "What is/was your latest occupational position at Volvo Cars". The text repsons was then coded as leadership position = "yes" if having any of the following positions; CEO, manager, director, executive, leader or foreman. Retirement during recession years was defined as years when Volvo due to economic downturns were forced to lay off large number of employees. These years where 2006, 2009 and 2013. The variable was dichotomized as retired during recession year (yes/no). Whether it was the person's own decision to retire was captured with one item created for this study: "Did you chose yourself to retire or did you feel pressured (by for example health, employer etc.): Have either retired or still working; if retired the response scale ranged response scale (1–5): 1 = "It was entirely my own decision to retire" to 5 = "I could not chose at all, I was forced to retire". Life events defined as having a severely ill close relative, measured with one item from a scale developed by Welin et al [29], "Have any of your closest relatives been severely ill or had a serious accident, and has this affected your decision to retire?", with a categorical response scale "no", "yes, and it affected my wish to retire" "yes, and it affected my wish to continue working.

Three models were calculated; first a crude model, a second model was adjusted for work variables frequency of heavy physical demands, leadership quality and having a leadership position and retirement during crisis years and if it was the person's own decision to retire. In the third model off-work factors were added: age difference to partner and life events. JDC and ERI were analysed separately. All missing items for demand, control, effort and reward were imputed accordingly: subjects with ≥50% missing items per variable were excluded. For subjects with <50% missing items per a variable, mean scores of the remaining items in each variable, were imputed on individual level.

## Results

In total, analyses were based on 1343 subjects (85.4% men), who had filled-in any psychosocial questions, and then further divided into subsamples for analyses using JDC (n = 1312) or ERI dimensions (n = 1307). In the final sample 42.6% were cases and 57.4% were controls, and 25% of women were cases versus 75% controls (Table 1).

Multiple logistic regression analyses between psychosocial variables in total-sample analyses (Table 2), revealed mostly small and non-significant effects. A few calculations illustrated surprising results, such as augmented odds ratios for remaining at work if experiencing imbalance between effort and reward. However, these results became non-significant in fully adjusted models.

Analyses conducted among blue-collar workers (Table 3), showed that high strain jobs meant notable lower odds ratios for continued work ≥63 years, in both unadjusted and fully

**Table 1. Cohort characteristics according to demographics and psychosocial variables.**

| | All | | Cases | | Controls | |
|---|---|---|---|---|---|---|
| **All, N (%)** | 1343 | | 572 | (42.6) | 771 | (57.4) |
| -Men | 1140 | | 521 | (45.7) | 619 | (54.3) |
| -Women | 196 | | 49 | (25.0) | 147 | (75.0) |
| Blue-collar, N (%) | 561 | | 236 | (42.1) | 325 | (57.9) |
| -Men | 458 | | 215 | (46.7) | 243 | (53.3) |
| -Women | 95 | | 19 | (19.8) | 77 | (80.2) |
| White-collar, N (%) | 782 | | 336 | (43.0) | 446 | (57.0) |
| -Men | 680 | | 306 | (45.0) | 374 | (55.0) |
| -Women | 100 | | 30 | (30.0) | 70 | (70.0) |
| Heavy physical demands, mean (sd) | 2.0 | (1.1) | 2.0 | (1.0) | 2.0 | 1.0 |
| Leadership quality, mean (sd) | 12.7 | (3.5) | 12.7 | (3.4) | 12.6 | 3.5 |
| Having a leadership position (yes), N (%) | 321 | (26.5) | 126 | (23.8) | 195 | (28.6) |
| Retired in recession years (yes), N (%) | 626 | (46.6) | 115 | (20.1) | 511 | (66.3) |
| Own decision to retire, mean (sd) | 2.7 | (1.3) | | | | |
| Age difference to partner | | | | | | |
| • Younger | 695 | (69.2) | 321 | (72.0) | 374 | (67.0) |
| • Same age | 106 | (10.6) | 51 | (11.4) | 55 | (9.9) |
| • Older | 203 | (20.2) | 74 | (16.6) | 129 | (23.1) |
| Life event affecting retirement (yes), N (%) | 174 | (13.3) | 81 | (14.4) | 93 | (12.4) |
| Participants experiencing high strain, N (%)* | | | | | | |
| All | 496 | (37.8) | 215 | (38.0) | 281 | (37.7) |
| -Men | 4108 | (36.3) | 196 | (37.8) | 212 | (34.9) |
| -Women | 88 | (47.1) | 19 | (39.6) | 69 | (49.6) |
| Blue-collar | 186 | (34.5) | 61 | (24.5) | 98 | (31.3) |
| -Men | 140 | (31.0) | 61 | (28.5) | 79 | (33.2) |
| -Women | 46 | (52.9) | 6 | (33.3) | 40 | (58.0) |
| White-collar | 235 | (30.4) | 116 | (34.7) | 119 | (27.1) |
| -Men | 203 | (30.2) | 105 | (34.5) | 98 | (26.6) |
| -Women | 32 | (32.0) | 11 | (36.7) | 21 | (30.0) |
| Participants with an ERI-ratio>1.0, N (%)** | | | | | | |
| All | 503 | (38.5) | 230 | (40.9) | 274 | (36.5) |
| -Men | 414 | (36.9) | 210 | (40.9) | 204 | (33.6) |
| -Women | 89 | (47.9) | 20 | (40.7) | 69 | (50.0) |
| Blue-collar | 208 | (39.0) | 88 | (38.3) | 120 | (39.5) |
| -Men | 168 | (37.5) | 81 | (38.2) | 87 | (36.9) |
| -Women | 40 | (46.5) | 7 | (38.9) | 33 | (48.5) |
| White-collar | 295 | (38.2) | 142 | (42.8) | 153 | (34.7) |
| -Men | 246 | (36.5) | 129 | (42.7) | 117 | (31.5) |
| -Women | 49 | (49.0) | 13 | (43.3) | 36 | (51.4) |

* based on n = 1312;

** based on n = 1307

adjusted models (OR 0.4; 95% CI 0.1–0.96. Active work was also associated to lowered OR of continued work, but results were non-significant. When stratifying by gender similar, but non-significant results were found among men. Few blue-collar women resulted in wide confidence intervals in fully adjusted analyses. Results for effort-reward imbalance mostly were

**Table 2. Regression analyses of psychosocial conditions between cases with extended work ≥63 years and controls who retired ≤62 years.**

| | | Crude | | Model 2 | | Model 3 | |
|---|---|---|---|---|---|---|---|
| | | *ratio* | *95% CI* | *ratio* | *95% CI* | *ratio* | *95% CI* |
| | | | *p-value* | | *p-value* | | *p-value* |
| **All** *n = 1312* | High strain | 1 | 0.7–1.4 | 1.1 | 0.8–1.7 | 0.9 | 0.5–1.5 |
| | | | 0.9 | | 0.5 | | 0.7 |
| | Active | 0.9 | 0.6–1.3 | 0.9 | 0.6–1.4 | 0.9 | 0.5–1.4 |
| | | | 0.5 | | 0.8 | | 0.6 |
| | Passive | 1.1 | 0.8–1.7 | 1.2 | 0.8–1.8 | 0.9 | 0.5–1.5 |
| | | | 0.4 | | 0.3 | | 0.7 |
| | Low strain | 1 | *Reference* | 1 | *Reference* | 1 | *Reference* |
| *n = 1307* | ERI ≥ 1.0 | 1.2 | 0.96–1.5 | **1.4** | **1.0–1.8** | 1.2 | 0.8–1.7 |
| | | | 0.1 | | **0.02** | | 0.7 |
| **Men** *n = 1125* | High strain | 1.2 | 0.8–1.7 | 0.9 | 0.5–1.4 | 1 | 0.6–1.7 |
| | | | 0.3 | | 0.6 | | 0.95 |
| | Active | 0.9 | 0.6–1.4 | 0.8 | 0.5–1.4 | 0.9 | 0.5–1.6 |
| | | | 0.7 | | 0.5 | | 0.8 |
| | Passive | 1.3 | 0-9-1.7 | 0.9 | 0.5–1.4 | 0.9 | 0.5–1.9 |
| | | | 0.2 | | 0.6 | | 0.8 |
| | Low strain | 1 | *Reference* | 1 | *Reference* | 1 | *Reference* |
| *n = 1121* | ERI ≥ 1.0 | **1.4** | **1.1–1.8** | 1.4 | 1.0–2.0 | 1.3 | 0.9–2.0 |
| | | | **0.01** | | 0.05 | | 0.2 |
| **Women** *n = 187* | High strain | 0.5 | 0.2–1.4 | 1.2 | 0.3–6.0 | 0.9 | 0.1–10.5 |
| | | | 0.2 | | 0.8 | | 0.4 |
| | Active | 0.5 | 0.1–1.8 | 0.8 | 0.1–4.6 | 0.4 | 0.03–4.1 |
| | | | 0.3 | | 0.8 | | 0.4 |
| | Passive | 0.8 | 0.3–2.3 0.6 | 1.8 | 0.4–8.6 | 0.7 | 0.01–7.8 |
| | | | | | 0.5 | | 0.8 |
| | Low strain | 1 | *Reference* | 1 | *Reference* | 1 | *Reference* |
| *n = 186* | ERI ≥ 1.0 | 0.7 | 0.4–1.4 | 0.7 | 0.2–2.3 | 1.1 | 0.2–5.0 |
| | | | 0.3 | | 0.6 | | 0.9 |

Model2: adjusted for frequency of heavy physical demands, leadership quality and having a leadership position and retirement during crisis years and if it was the person's own decision to retire

Model3: model 2 + agedifference to partner and life event

small and non-significant, except for analyses in women which displayed considerably lowered odds ratios, but due to few women, confidence interval were wide.

Regression analyses in white collar-workers (Table 4) displayed several unexpected results as high strain was related to considerably increased odds for continued work in older ages for total sample and for men, but the effects did not persist in fully adjusted models. In total sample and male white-collar workers high imbalance between efforts and just reward at work meant increased odds for working ≥ 63 years of age, but these effects became non-significant when adjusting for all potential covariates. Among white-collar women both high strain and ERI illustrated increased odds of continued work but confidence intervals were wide and non-significant.

## Discussion

This case-control study found inconsistent results regarding relationships between psychosocial factors, measured by the JDC- and ERI-model, and extended work ≥ 63 years. In total

**Table 3. Regression analyses of psychosocial conditions between cases with extended work ≥63 years and controls who retired ≤62 years in *blue-collar* workers.**

| | | Crude | | | Model 2 | | | Model 3 | | |
|---|---|---|---|---|---|---|---|---|---|---|
| | | *ratio* | *95% CI* *p-value* | | *ratio* | *95% CI* *p-value* | | *ratio* | *95% CI* *p-value* | |
| **All** n = 539 | High strain | **0.5** | **0.3–0.9** | **0.01** | **0.4** | **0.3–0.9** | **0.03** | **0.4** | **0.1–0.96** | **0.04** |
| | Active | 0.6 | 0.4–1.1 | 0.08 | 0.7 | 0.4–1.2 | 0.2 | 0.7 | 0.3–1.7 | 0.4 |
| | Passive | 1 | 0.6–1.7 | 0.99 | 1.2 | 0.7–2.2 | 0.5 | 1.3 | 0.4–3.7 | 0.6 |
| | Low strain | 1 | *Reference* | | 1 | *Reference* | | 1 | *Reference* | |
| n = 534 | ERI ≥ 1.0 | 0.95 | 0.7–1.4 | 0.8 | 1 | 0.6–1.9 | 0.9 | 0.8 | 0.4–1.7 | 0.3 |
| **Men** n = 452 | High strain | 0.7 | 0.4–1.2 | 0.2 | 0.7 | 0.4–1.3 | 0.3 | 0.5 | 0.2–1.6 | 0.3 |
| | Active | 0.7 | 0.4–1.2 | 0.2 | 0.7 | 0.4–1.3 | 0.3 | 0.8 | 0.3–1.9 | 0.6 |
| | Passive | 1.2 | 0.7–2.2 | 0.5 | 1.5 | 0.8–3.0 | 0.2 | 1.9 | 0.6–5.9 | 0.3 |
| | Low strain | 1 | *Reference* | | 1 | *Reference* | | 1 | *Reference* | |
| n = 448 | ERI ≥ 1.0 | 1.1 | 0.7–1.6 | 0.8 | 1.3 | 0.7–2.4 | 0.5 | 1 | 0.5–2.3 | 0.97 |
| **Women** N = 87 | High strain | 0.2 | 0.01–2.8 | 0.08 | N/A | <0.001–0.4 | 0.01 | N/A | | 0 |
| | Active | 0.3 | 0.03–2.9 | 0.3 | 0.1 | 0.02–1.8 | 0.1 | N/A | | |
| | Passive | 0.4 | 0.044–3.8 | 0.4 | 0.1 | 0.03–1.8 | 0.08 | N/A | | |
| | Low strain | 1 | *Reference* | | 1 | *Reference* | | 1 | *Reference* | |
| n = 86 | ERI ≥ 1.0 | 0.7 | 0.2–1.9 | 0.5 | N/A | | | 0.3 | 0.04–2.1 | 0.2 |

Model2: adjusted for social support, retirement during economic crisis years

Model3: model 2 + civil status, previous diseases, life events, stress at home

sample and gender stratified analyses no JDC-variables were significantly related to retirement in models included all potential covariates. When stratifying by blue- or white-collar work it was found that, for blue-collar workers high strain was related to retirement at 62 or earlier in all models. When stratifying by gender similar, but non-significant results were found for the blue-collar men. Surprisingly, for white-collar workers high strain was related to notably increased odds ratios for continuing to work after 62 years, but these results did not persist in fully adjusted models. When analysing white-collar men and women separately it was found that high strain was related to continuing to work after 62 years in men. Similar contradictive results were found for imbalance between effort and reward, where high ERI was associated with increased odds for continuing to work for all white-collar workers, and for white-collar men, but these results were non-significant in the fully adjusted models.

Previous research have found that job strain is associated with intentions to leave work early [20], although gender and sector specific differences have been reported [11,12]. Gender stratified analyses in the current study did not support that high strain was related to earlier

**Table 4. Regression analyses of psychosocial conditions between cases with extended work ≥63 years and controls who retired ≤62 years in *white-collar* workers.**

| | | Crude | | Model 2* | | Model 3** | |
|---|---|---|---|---|---|---|---|
| | | ratio | 95% CI | ratio | 95% CI | ratio | 95% CI |
| | | | p-value | | p-value | | p-value |
| **All** | High strain | **1.6** | **1.0–2.3** | 1.3 | 0.8–2.2 | 1.4 | 0.8–2.6 |
| | | | **0.04** | | 0.3 | | 0.2 |
| | Active | 1.1 | 0.7–1.6 | 1 | 0.6–1.7 | 1.1 | 0.6–1.9 |
| | | | 0.8 | | 0.9 | | 0.8 |
| | Passive | 1.2 | 0.8–1.9 | 1 | 0.6–1.8 | 1.1 | 0.6–2.0 |
| | | | 0.4 | | 0.9 | | 0.9 |
| | Low strain | 1 | Reference | 1 | Reference | 1 | Reference |
| n = 773 | ERI ≥ 1.0 | **1.4** | **1.1–1.9** | 1.4 | 0.9–2.1 | 1.3 | 0.8–2.1 |
| | | | **0** | | 0.1 | | 0.2 |
| **Men** n = 673 | High strain | **1.7** | **1.1–2.6** | 1.4 | 0.8–2.3 | 1.4 | 0.8–2.7 |
| | | | **0.02** | | 0.3 | | 0.2 |
| | Active | 1.2 | 0.8–1.8 | 1.1 | 0.8–2.4 | 1.2 | 0.6–2.2 |
| | | | 0.5 | | 0.8 | | 0.6 |
| | Passive | 1.3 | 0.8–2.2 | 1 | 0.6–1.9 | 1.1 | 0.6–2.1 |
| | | | 0.3 | | 0.9 | | 0.8 |
| | Low strain | 1 | Reference | 1 | Reference | 1 | Reference |
| n = 673 | ERI ≥ 1.0 | **1.6** | **1.2–2.2** | 1.5 | 0.97–2.3 | 1.4 | 0.9–2.3 |
| | | | **0.003** | | 0.06 | | 0.2 |
| **Women** | High strain | 0.9 | 0.3–2.9 | 1.8 | 0.3–9.3 | 1.4 | 0.1–14.2 |
| | | | 0.9 | | 0.5 | | 0.8 |
| | Active | 0.5 | 0.1–1.8 | 0.5 | 0.1–4.2 | 0.4 | 0.03–4.4 |
| | | | 0.3 | | 0.6 | | 0.4 |
| | Passive | 0.7 | 0.2–2.4 | 1.8 | 0.3–10.0 | 0.9 | 0.1–9.5 |
| | | | 0.5 | | 0.5 | | 0.9 |
| | Low strain | 1 | Reference | 1 | Reference | 1 | Reference |
| n = 100 | ERI ≥ 1.0 | 0.7 | 0.3–1.7 | 1.7 | 0.4–7.5 | 2.3 | 0.3–15.5 |
| | | | 0.5 | | 0.4 | | 0.4 |

Model2: adjusted for social support, retirement during economic crisis years

Model3: model 2 + civil status, previous diseases, life events, stress at home

retirement for women. However, when stratifying by blue- or white-collar sector, high strain was related to retirement before 63 years for blue-collar workers. Our findings are similar to the results from a previous study [11], that found a relationship between job demands and early retirement intention, although in both blue- and white-collar workers. Blue-collar workers in the car manufacturing industry have a different working situation compared to white-collar workers with more physically demanding and strenuous working conditions [8]. Moreover, blue-collar workers generally tend to retire at an earlier age compared to white-collar workers [9]. It is therefore plausible that the process that drive retirement intention work differently for the two groups. Some demands may be more important for blue-collar workers and others may be more relevant to white-collar workers when predicting retirement intention [11].

A major limitation of this study is the risk of recall bias, as some of the participants retired up to ten years prior to completing the questionnaire. Indeed, the difficulty in assessing past exposures and the risk of recall bias is a well-known problem in case-control studies [30],

where the estmiate of exposure is usually based on past records or recall of past events by participants. The accuracy of recall in interviews has been studied for a number of exposures with varying results [31]. It has been suggested that bias or imprecision in estimated exposure can be difficult to avoid in case-control studies, and therefore it is important to consider if these errors are likely to differ between cases and controls [31]. In order to assess the magnitude of this problem we performed sensitivity analyses and investigated differences between the psychosocial variables (job demand, job control and the ERI-ratio) depending on the time lag between the survey and retirement. The time since filling in the survey was categorized as currently working, or retired since 1 years, 2–4 years or more than 5 years and analysed using two-way ANOVA, with a p-value of 0.05 as significance level. Analyses illustrated that there were no statistically significant differences in the examined psychosocial variables, between those currently working or leaving work recently or more than 5 years ago, regardless of total-sample analyses or when stratifying by cases or controls. Nevertheless, recall bias can be a threat to the validity of the study. Only subjective measures of working conditions were used in the study as the main focus was the psychosocial working environment. In the current study, the JDC model was used to assess demands and this model was developed when studying blue-collar working conditions [16] and is therefore suitable to assess the job demands of this group. However, it may not be the best model to capture some of the demands faced by white-collar workers such as cognitive demands. Moreover, there was a small sample of women, thus the results relating to gender comparisons need to be treated with caution due to the limited statistical power. Further limitations included the fact that health status just before retirement, and individual differences (for example coping) were not measured. These factors could have an influence on the decision to retire.

This study can have practical implications because it contains data from the largest car manufacturer in Sweden, stratifies for blue- and white-collar work and gender, and includes important confounders such as life events. The results indicate that demanding psychosocial working conditions are related to earlier retirement for blue-collar workers. Hence, it is important to promote sustainable working conditions for these groups in order to retain older workers. For example, it may be useful to examine the job design to investigate if it is possible to decrease demands and provide more job autonomy for blue-collar workers. However, this case-control study investigates associations and not causation, and it is important to note that prospective studies will be required to determine what job factors that have a causal role in the timing of retirement.

## Conclusions

In conclusion, this study found that for blue-collar workers high strain was related to retirement before 63 years.

## Supporting information

**S1 Data.**
(DOCX)

## Acknowledgments

Volvo Cars

Adnan Baloch, statistician at Department of Occupational and Environmental Medicine, Sahlgrenska Academy and University of Gothenburg, Gothenburg, Sweden.

## Author Contributions

**Conceptualization:** Kristina Gyllensten, Kjell Torén, Mats Hagberg, Mia Söderberg.

**Data curation:** Kjell Torén, Mats Hagberg.

**Formal analysis:** Kristina Gyllensten, Kjell Torén, Mats Hagberg, Mia Söderberg.

**Funding acquisition:** Kjell Torén, Mats Hagberg.

**Investigation:** Kristina Gyllensten, Kjell Torén, Mats Hagberg, Mia Söderberg.

**Methodology:** Kristina Gyllensten, Kjell Torén, Mats Hagberg, Mia Söderberg.

**Project administration:** Kristina Gyllensten, Kjell Torén, Mats Hagberg, Mia Söderberg.

**Supervision:** Kjell Torén, Mats Hagberg.

**Writing – original draft:** Kristina Gyllensten, Kjell Torén, Mats Hagberg, Mia Söderberg.

**Writing – review & editing:** Kristina Gyllensten, Kjell Torén, Mats Hagberg, Mia Söderberg.

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
