## [Decision Letter · Decision Letter 0]

18 Aug 2019

PONE-D-19-15489

A sustainable working life in the car manufacturing industry: The role of psychosocial factors, gender and occupation

PLOS ONE

Dear Dr Gyllensten,

Thank you for submitting your manuscript to PLOS ONE. After careful consideration, we feel that it has merit but does not fully meet PLOS ONE’s publication criteria as it currently stands. Therefore, we invite you to submit a revised version of the manuscript that addresses the points raised during the review process.

Below, you will see that three experts in the field have reviewed your paper. Overall, two of them stated it has a certain potential, but more work is needed before considering its publication in PLoS ONE. As -in my view- most of the appended comments and issues are amendable, I would like to give the authors the chance to address and respond all the comments provided by our reviewers. Although all the comments should be properly addressed, please pay special attention to those related to the methods, data analysis, scope and limitations of the study,

We would appreciate receiving your revised manuscript by Oct 02 2019 11:59PM. To enhance the reproducibility of your results, we recommend that if applicable you deposit your laboratory protocols in protocols.io, where a protocol can be assigned its own identifier (DOI) such that it can be cited independently in the future. For instructions see: http://journals.plos.org/plosone/s/submission-guidelines#loc-laboratory-protocols

We look forward to receiving your revised manuscript.

Kind regards,

Sergio A. Useche, Ph.D.

Academic Editor

PLOS ONE

Journal Requirements:

1. Please include a copy of any survey questions or questionnaire used in the study that have not been published previously, in both the original language and English, as Supporting Information. Please ensure that you have provided citations for any scales published previously.

2. We note that you have included in the following statement in your Acknowledgements "Volo Cars [sic]". We feel that the involvement of Volvo Cars should be included in the Competing Interests statement. Please state what role Volvo had in  conducting the study and decision to publish. Please include an updated Competing Interests statement in your Cover Letter and we will update the submission form on your behalf.

3. Thank you for including your financial disclosure statement; "The study was partly funded by AgeCap - a centre for aging and health at Gothenburg University. The funder did not have any involvement in the study, in the writing of the manuscript or the decision to submit the paper for publication.  "

4. Thank you for including your ethics statement: "The study was approved by the Ethical Review Board in Gothenburg in Sweden, dnr 371-15. The consent was written."

6. We note you have included a table to which you do not refer in the text of your manuscript. Please ensure that you refer to Tables 1 & 4 in your text; if accepted, production will need this reference to link the reader to the Table.

Reviewers' comments:

Reviewer's Responses to Questions

**Comments to the Author**

1. Is the manuscript technically sound, and do the data support the conclusions?

Reviewer #1: Partly

Reviewer #2: Partly

Reviewer #3: No

2. Has the statistical analysis been performed appropriately and rigorously? 

Reviewer #1: No

Reviewer #2: No

Reviewer #3: Yes

3. Have the authors made all data underlying the findings in their manuscript fully available?

Reviewer #1: Yes

Reviewer #2: Yes

Reviewer #3: Yes

4. Is the manuscript presented in an intelligible fashion and written in standard English?

Reviewer #1: Yes

Reviewer #2: Yes

Reviewer #3: No

5. Review Comments to the Author

Reviewer #1: This study uses state of the art measures of stress at work – job strain (JDC) and effort-reward imbalance (ERI) – to evaluate the influence of stress at work on early retirement (age 55-62) vs continuing to work age >63. The findings of high strain being related to early retirement in blue collar workers but both high strain and high ERI being related to continued work in white collar men are interesting and of potential importance with respect to reducing early retirement in blue collar workers (assuming that’s considered a desirable goal). There are some concerns that need the authors’ attention.

1. Rather than, or in addition to, covarying several variables, there are some – especially previous diseases, social support at work, and stress at home – that, rather than being confounders, could be moderators (which can be documented by testing their interactions with job strain and/or EFI) of the influence of JDC and/or ERI on retirement time. If blue collar workers with previous disease, for example, are more likely to retire early if they are in high strain jobs, they could be targeted for interventions that might help them cope better and remain at work. If those with high support at work are less likely to retire early even if in high strain jobs, interventions to increase support at work could help workers in high strain jobs continue to work longer.

2. It would also be good to test the interaction of blue x white collar x JDC/ERI, to clearly document that effects of JDC and ERI on time of retirement are moderated by blue/white collar status.

3. There is no mention in the text that I can find of Table 1, which could include whether Cases and Controls differ significantly on any of the characteristics – e.g., 25% of women being cases vs 75% controls.

4. While the ratios and 95% CIs do document effects that are statistically significant, it would be helpful if they also included p-values for these effects.

5. The authors do note that a major limitation of this study is the risk of recall bias – e.g., those blue collar workers who retired before age 63 may be more likely to recall their job as having high strain than those who work past age 63. The authors need to note that before conclusions can be drawn regarding causality, the effect of high strain to increase incidence of early retirement in blue collar workers and of high strain and ERI delaying retirement in white collar workers, it will be essential to do a prospective study of current workers to determine whether high strain predicts early retirement in blue collar workers and high strain and ERI predict later retirement in which collar men before any conclusions can be drawn (and interventions developed and evaluated) regarding causality. If the JDC and ERI assessments were done while any of the workers in the current study were still working, it could be possible to see if JDC or ERI levels predict retirement ages in them.

Reviewer #2: 1. Is the manuscript technically sound, and do the data support the conclusions?

The most important limitation of this study is that psychosocial working conditions is measured after the decision to retire or not, and often many years later. This means that one may primarily measure the respondent’s justification for his/her decision and not how he/she felt about the working conditions in the time leading up to the decision. There is a lot of psychological research on cognitive dissonance and similar phenomena that give rise to considerable skepticism about the validity of such data.

Another limitation, although one shared with most studies in this tradition, is that only subjective measures of working conditions are used. This should, however, be clearly acknowledged. The research cannot address the issue of whether the working conditions in an objective sense has any effects or not.

Even beyond this, it should be made clear that findings are only descriptive, above all in the Aims section. Causal language should be avoided throughout the paper.

2. Has the statistical analysis been performed appropriately and rigorously?

A better justification of the selection of covariates is needed. Why are these particular covariates included? What about potential confounders not included?

I find several of the covariates problematic. What are, for instance, the implications of including health related variables that are likely to be endogenous to working conditions?

On the other hand it is notable that there are no measures of working and employment conditions apart from JDC and ERI. This means that the effects of these specific variables cannot be singled out. One illustration of this problem is provided by the descriptive results on distinguishing between management and other job positions (line 325ff.). Indeed, it seems strange to me that that this variable is not included as a covariate. Another obvious confounder (probably not measured) is the wage level.

Statistical power may be problematic (particularly in analyses of women), and should be addressed.

3. Have the authors made all data underlying the findings in their manuscript fully available?

No further comments.

4. Is the manuscript presented in an intelligible fashion and written in standard English?

No further comments.

Reviewer #3: This manuscript describes a study examining the association between psychosocial work characteristics and the propensity to retire early versus continue to work past the age of 62. The results are thought-provoking, indicating associations in different directions for blue collar male and white collar male car manufacturing workers.

However, the research design is somewhat problematic (and a little unclear). All employees who worked at a particular car company during the years 2005-2015, and either retired during this time (after age 55) or continued working past the age of 62 were included in the sample. The first potentially problematic issue is that people who were 55 years or older and who were laid off during this time were included in the sample (I think). Since the study is examining employees’ choices about retirement, it seems more appropriate to exclude the employees who did not have a choice.

The second problem is the timing of the data collection. Some of the participants may have retired 10 years prior to the survey data collection. Much literature suggests that recall bias can be a very real threat to validity--- that a person’s current state can greatly influence his or her report of previous exposures. While the authors attempt to dispel the seriousness of this potential threat, I find myself unconvinced.

Thirdly, the authors are not very clear as to the time period that participants were asked to consider when answering the psychosocial work characteristics questions. Were they asked to consider the work conditions being experienced at the time of their retirement? And what about those still working? Were they asked to report on the conditions at the time when they could have retired? The stress at home question seems to have asked about the last 5 years which could have been substantially post-retirement for some. To sum, a retrospective study design lends itself to several potential threats to validity.

In terms of the sample, the number of women is quite small. When the stratified regressions are conducted, there are very few in each of the categories of the psychosocial work characteristics variables. The study might be better served by only including men, and thus not making gender comparisons.

While the authors briefly review the associations between psychosocial working conditions and employee health, they do not examine the potential relationships between retiring at a later age and employee health. What if retiring at a later age is good for employers but bad for employee health? This topic needs to be addressed in the introduction.

Lastly, the discussion section does not clearly explain how this study moves the field forward. What do we know after reading this study that we didn’t know before? This study seems to be another flawed study that adds to the inconsistencies in the previous research. The same possible mechanisms/explanations for the effects were discernable from previous research.

6. PLOS authors have the option to publish the peer review history of their article (what does this mean?). If published, this will include your full peer review and any attached files.

Reviewer #1: No

Reviewer #2: No

Reviewer #3: No

---

## [Author Response · Author response to Decision Letter 0]

5 Nov 2019

Note: Page numbers refer to the pages on the revised manuscript with track changes. 

Editors comments: 

Comment 1. A document has been included with supporting information including survey questions that have not been published previously. 

In the methods section on page 7 citations for scales published previously have been included. 

Comment 2. Updated competing interest statement. Thank you for updating the submission form on our behalf. 

Volvo Cars provided contact details, age, details of level of employment and time of employment, of all employees that had worked/ and or retired at Volvo between the years of 2005-2015 that were 55 years or older. Volvo did not have any further involvement in conducting the study or in the decision to publish the study. 

Comment 3. The funding statement has been updated.

The study was funded by AgeCap – a centre for aging and health at Gothenburg University and by the Swedish Research Council for Health, Working life and Wellfare. The funders did not have any involvement in the study, the writing of the manuscript or the decision to submit the paper for publication. There was no additional external funding received for this study. 

Comment 4. The full name of the ethics committee has been included in the methods section on page 6, and in the submission form. 

Comment 5. Data available on request.

a) Data cannot be shared publicly, but, anonymised data is available with an approval from an ethical review board. For data requests, contact: Department of occupational and environmental medicine, Gothenburg University, Box 414, 405 30, Gothenburg, Sweden. 

Comment 6. Table 1 and 4 is now referred to in the text in the results section on page 10 and 14. 

Reviewer 1 

Comment 1. We agree that mentioned analyses may add more information to the decision of retiring. Exploratory analyses were conducted testing interaction effects between the main psychosocial variables (job demand-control / effort-reward imbalance) and the suggested variables (previous diseases, social support at work). The interactive effects were mostly small and all were non-significant. We conclude that these variables are not moderators, and will not add the results to our paper.

Comment 2. Exploratory analyses showed interactive effects between high strain and blue-/white-collar worker. However, the paper already includes analyses stratified by blue- white-collar worker, and illustrates effects of psychosocial variables may differ depending on being blue- or white-collar workers. 

Comment 3. Table 1 is now mentioned in the text on page 10. 

Comment 4. P-values are included in table 2-4 in the manuscript on pages 12, 13, 14. Some calculation errors were noted and corrected. Our results remain largely similar

Comment 5. Conclusions regarding causality has been removed on the following pages. 

In the abstract on page 2. 

In the aims on page 5. 

In the implications on page 21. 

Reviewer 2 

Comment 1. In limitations on page 20 a sentence has been added that acknowledges that only subjective measures of working conditions were used in the study. 

Casual language has been removed in the following sections: 

In the abstract on page 2. 

In the aims on page 5. 

In the implications on page 21. 

Comment 2. We evaluated several confounders we identified as important, based on previous literature and meetings with Volvo staff (including mangers, white-collar and blue-collar workers). The following variables were considered: civil status, age of partner in relation to the participant, country of birth, retirement during a year of recession, being offered a beneficial retirement deal, if retirement was the persons own decision, previous diseases, work ability at the time of retirement, social support at work, need for recovery after work at the time of retirement and stress at home at the time of retirement. Since working with a fairly small cohort, we performed stepwise purposeful selection according to Lemeshow & Hosmer (2000) reference 26, to slim our models, which resulted in the confounders presented in the paper.

We had access to other work condition variables (e.g. noise, physical demands), but our main aim with this paper was to focus on psychosocial variables. Also, this paper examines both white- and blue-collar Volvo workers. Noise and physical demands likely have little effect among white-collar workers. We plan to examine effects from physical work factors among blue-collar workers in future studies.

As suggested we examined other psychosocial variables that might be important such as managerial/executive position and overtime per week. Both variables had little effect and did not full-fill the criteria according to Lemeshow & Hosmer (2000). 

We did not have access to wage level, which is a limitation, but Volvo workers at this time, where provided with a beneficial pension deal, so that pension would be similar regarding if retiring at 62 or statutory Swedish pension age 65. Wage may therefore be of less importance in this cohort, than other similar studies.

Comment 3. The limited statistical power realting to the the small sample of women is addressed in the limitations on page 20. 

Reviewer 3 

Comment 1. The reviewer highlighted that people over 55 years that were laid off appeared to be included in the sample. Individuals who had been laid off were not included in the sample. Thank you for highlighting that this was unclear. A paragraph clarifying this has been included in the study population section on page 6. 

Comment 2. We agree that a major limitation of this study is the risk of recall bias. In the manuscript we have revised the text relating to limitations on page 20 to further acknowledge this problem. 

Comment 3. The reviewer highlighted that the time period that the participants were asked to consider their psychosocial environment was unclear. The cases were asked to consider their current working environment, or the working environment at their last workplace at Volvo. The controls were asked to consider the working environment at their last workplace at Volvo. Information about this has been added in the section describing the questionnaire on page 7. 

Comment 4. We agree that sample of women is quite small and the problem with power has been acknowledged in the section on limitations on page 20. However, despite the small sample we still think there is a value with including women in the analysis. 

Comment 5. A recognition of the fact that late retirement can be problematic for certain employees have been added in the beginning of the introduction on page 3. 

Comment 6. There is currently an active debate, especially regarding job demand-control, that these models that were developed several decades ago, and due to a changing labor market, may have different meaning compared to when they were created. During the time of the forming of the models, the typical high strain work situation was found in blue-collar workers e.g. in line production. Many, however, argue that due to technology, several white-collar workers in modern labor market have increasing demands linked to less boundaries and longer work hours. The understanding how to interpret and measure psychosocial work conditions needs to be updated, but to do so, there is a need to first establish limitations in standard methods.

---

## [Decision Letter · Decision Letter 1]

4 Dec 2019

PONE-D-19-15489R1

A sustainable working life in the car manufacturing industry: The role of psychosocial factors, gender and occupation

PLOS ONE

Dear Dr Gyllensten,

Thank you for submitting your manuscript to PLOS ONE. After careful consideration, we feel that it has merit but does not fully meet PLOS ONE’s publication criteria as it currently stands. Therefore, we invite you to submit a revised version of the manuscript that addresses the points raised during the review process.

The paper has been reviewed for a second time by two acknowledged experts in the topic addressed in your manuscript. Although some improvements are evident, it is true (especially considering the feedback received from Reviewer # 2) that more precision, extension and efforts are needed in regard to the amendments they asked for during their first review.

Actually, and considering the aforementioned, one of the reviewers have advised the rejection of the paper. However, I believe the manuscript has potential, and the topic is worth of investigation; thus, I would like to encourage the authors to perform an adequate revision of it, involving all the comments received in both review phases, with more level of detail and accuracy in regard to what is expected from reviewers. Please carefully follow the journal guidelines and cover all the pertinent queries.

We would appreciate receiving your revised manuscript by Jan 18 2020 11:59PM. To enhance the reproducibility of your results, we recommend that if applicable you deposit your laboratory protocols in protocols.io, where a protocol can be assigned its own identifier (DOI) such that it can be cited independently in the future. For instructions see: http://journals.plos.org/plosone/s/submission-guidelines#loc-laboratory-protocols

We look forward to receiving your revised manuscript.

Kind regards,

Sergio A. Useche, Ph.D.

Academic Editor

PLOS ONE

Reviewers' comments:

Reviewer's Responses to Questions

**Comments to the Author**

1. If the authors have adequately addressed your comments raised in a previous round of review and you feel that this manuscript is now acceptable for publication, you may indicate that here to bypass the “Comments to the Author” section, enter your conflict of interest statement in the “Confidential to Editor” section, and submit your "Accept" recommendation.

Reviewer #1: (No Response)

Reviewer #2: (No Response)

2. Is the manuscript technically sound, and do the data support the conclusions?

Reviewer #1: Partly

Reviewer #2: Partly

3. Has the statistical analysis been performed appropriately and rigorously? 

Reviewer #1: Yes

Reviewer #2: No

4. Have the authors made all data underlying the findings in their manuscript fully available?

Reviewer #1: Yes

Reviewer #2: Yes

5. Is the manuscript presented in an intelligible fashion and written in standard English?

Reviewer #1: Yes

Reviewer #2: Yes

6. Review Comments to the Author

Reviewer #1: The authors have done a good job of responding to my concerns in this revision. That said, there remain some concerns that need to be addressed.

1. The exploratory analyses they performed in response to my Comment 1 showed no significant interactions between job variables and other psychosocial variables (previous diseases, social support at work, etc.) and hence support the conclusion that effects of job characteristics on retirement time were not moderated by these other variables. Rather than NOT adding these results to the paper, I believe it would be better to report these non-significant interactions in the paper.

2. Similarly, I would include in the revised ms the results of the exploratory analyses that showed significant interactions between high strain and blue/white collar status – thereby strengthening the case that blue/white collar status moderates the influence of job factors on retirement time.

3. The changes in text they have made to reduce conclusions regarding implications of their findings for causality of early retirement are good. In addition, I believe it would be good to indicate at the end of the Discussion that prospective studies will be required to document further that job factors are playing a causal role in retirement time.

4. A minor point: They need to insert “p-value” beneath “95% CI” at the top of the columns in Table 4.

Reviewer #2: Unfortunately, I do not find much improvement in this version. Although a few sentences are changed, the whole justification for the paper set up in the Introduction is the need for knowledge about causes (a few examples, 52-53, 62-64, 101-102). Im the paper is not intended to address this, then what is the contribution? I do not find any clear statement of that.

With regard to recall bias, problems of confounding, etc. there is no concrete and serious discussion of this might affect the results of the analyses.

7. PLOS authors have the option to publish the peer review history of their article (what does this mean?). If published, this will include your full peer review and any attached files.

Reviewer #1: No

Reviewer #2: No

---

## [Author Response · Author response to Decision Letter 1]

27 Feb 2020

Reviewer #1

1 & 2. We agree to these well-thought of and insightful comments, but after thorough discussion in our project group we have concluded the following. It’s correct that several of the chosen covariates could possible also be moderators. We do, however, believe that there are many studies in this field with a similar design examining psychosocial variables and using models with covariates that could be both moderators or covariates. Even though it may add interesting information, is not common to calculate or display results on interaction effects. We still carried out exploratory analyses evaluating interaction, but none were significant. In the previous review we had other models, but as suggested from reviewer 2 we included more work related covariates and interaction effects between for example psychosocial variables and socioeconomic status are now not significant anymore. We agree to your suggestion that it may benefit the study in some ways, but we have decided not to expand our results with interaction analyses, since most studies do not and since interaction effects were non-significant it may not contribute that much. 

3. Thank you for this helpful suggestion. We have added a sentence about the fact that prospective studies are needed in order to find out more about casual factors. Page 19. 

4. Thank you, p-value has been added to Table 4. Page 14. 

Reviewer #2

Contribution of the paper 

We acknowledge that this study cannot discuss causes of retirement. However, we believe that this study adds to existing knowledge regarding the associations between psychosocial factors and the timing of retirement. There is little research investigating these factors in the car manufacturing industry. We have revised the aim in order to emphasize that we are investigating the relationships between psychosocial work factors and retirement, and thereby aim to add to existing knowledge. Page 5. 

Problems with recall bias 

We agree that this can be a serious problem, and have added further reasoning regarding this issue in case-control studies in the discussion. Page 17-18. 

Problems with confounding

We agree that our models needed to be revised and more covariates should be considered. Since modelling and used covariate is one of the main issues in the reviewer comments we have expanded the methodology section in which we explain the methodology for inclusion of covariates: stepwise purposeful selection as proposed by Hosmer & Lemeshow [1] (page 8). The rationale for using this method is that theoretically a broad variety of available covariates could be important in analyses in psychosocial work environment and retirement. However, since our cohort size is fairly small, and include analyses that are stratified by gender or blue-/white-collar workers, we wish to slim the models by only including covariates relevant for our analyses. Meaning, if a variable does not fulfill the criteria in the established selection process, they have little relevance for the model and evaluated effects.

The following potential confounders, which previous have been identified in the literature or by practical experience as important in the retirement process, were tested with stepwise purposeful selection as proposed by Hosmer & Lemeshow [1]:

civil status, age of partner in relation to the participant, country of birth, retirement during a year of recession, being offered a beneficial retirement deal, if retirement was the persons own decision, amount of years since retirement, previous diseases, work ability at the time of retirement, need for recovery after work, life events, stress at home, physical job index, quality of leadership, managerial position and social support at work. 

Per suggestion we especially added work related covariates such as frequency of physically heavy job tasks [2], leadership quality [3] and leadership position in the company (yes/no). The reviewer suggest salary as an important co-variate, unfortunately we do not have access to this variable. It has been suggested that for those with high status jobs, the salary per se is not determining for retirement, but rather social status and identity tied to the position. For blue-collar workers physical health and actual retirement benefits, is more important. If health if failing, salary may have less important. Furthermore, this cohort is slightly unique as the employees, due to a very generous deal with Volvo Cars, could retire at 62 years, and receive similar pension as if they would have retired at 65 years, which was the statutory retirement age during our observation period. Thus, salary hopefully, have less importance for in this particular cohort. The reviewer also suggested we remove health as a covariate given its endogenous property in relation to work conditions. We did include it in our purposeful stepwise confounding testing, and as it now, including all new work related variables, the variable previous diseases, did not fulfil the inclusion criteria, it is no longer included in our models. 

1. Lemeshow S, Hosmer D: Applied Logistic Regression (Wiley Series in Probability and Statistics. Hoboken: Wiley-Interscience; 2 Sub edition; 2000.

2. Arbetsmiljöverket: Arbetsmiljön 2013. In: Arbetsmiljöstatistisk Rapport 2014:3. vol. 2014:3; 2014.

3. Kristensen TS, Hannerz H, Høgh A, Borg V: The Copenhagen Psychosocial Questionnaire-a tool for the assessment and improvement of the psychosocial work environment. Scandinavian journal of work, environment & health 2005:438-449.

---

## [Decision Letter · Decision Letter 2]

5 Mar 2020

PONE-D-19-15489R2

A sustainable working life in the car manufacturing industry: The role of psychosocial factors, gender and occupation

PLOS ONE

Dear Dr Gyllensten,

Thank you for submitting your manuscript to PLOS ONE. After careful consideration, we feel that it has merit but does not fully meet PLOS ONE’s publication criteria as it currently stands. Therefore, we invite you to submit a revised version of the manuscript that addresses the points raised during the review process.

Thanks for the work done in regard to the previous set of revisions requested by our reviewers. Below, you will find an additional set of comments (most of them considerably minor, but still important) raised by the Reviewer #1. Please address all them with all the possible rigor for submitting the paper to a final round of reviews, and the potential acceptance of the paper in case the changes and rationales were accepted.

We would appreciate receiving your revised manuscript by Apr 19 2020 11:59PM. To enhance the reproducibility of your results, we recommend that if applicable you deposit your laboratory protocols in protocols.io, where a protocol can be assigned its own identifier (DOI) such that it can be cited independently in the future. For instructions see: http://journals.plos.org/plosone/s/submission-guidelines#loc-laboratory-protocols

We look forward to receiving your revised manuscript.

Kind regards,

Sergio A. Useche, Ph.D.

Academic Editor

PLOS ONE

Reviewers' comments:

Reviewer's Responses to Questions

**Comments to the Author**

1. If the authors have adequately addressed your comments raised in a previous round of review and you feel that this manuscript is now acceptable for publication, you may indicate that here to bypass the “Comments to the Author” section, enter your conflict of interest statement in the “Confidential to Editor” section, and submit your "Accept" recommendation.

Reviewer #1: (No Response)

2. Is the manuscript technically sound, and do the data support the conclusions?

Reviewer #1: Partly

3. Has the statistical analysis been performed appropriately and rigorously? 

Reviewer #1: Yes

4. Have the authors made all data underlying the findings in their manuscript fully available?

Reviewer #1: Yes

5. Is the manuscript presented in an intelligible fashion and written in standard English?

Reviewer #1: Yes

6. Review Comments to the Author

Reviewer #1: The authors have responded appropriately to my concerns 3 and 4. I have ongoing concerns with their argument for not reporting interactions.

Concern 1. Reporting nonsignificant interactions between job variables and JDS and ERI – the authors did assess these interactions and found that “none were significant.” This indicates that none of these job variables moderated effects of JDS and ERI on retirement, and I see no reason not to report this absence.

Concern 2. In response to my prior review they indicated they had done exploratory analyses that did show significant interactions between high strain and blue/white collar status. It is important to report this interaction, because it justifies doing separate analyses of JDS and ERI associations with retirement time in blue- and white-collar workers.

In the text regarding the regression analyses shown in Table 4 for white-collar workers, they neglect to report that, in addition to high strain being associated with extended work in all and male workers, high ERI is also associated with extended work in all and male subjects -- ratios 1.4 (1.1-1.9) and 1.7 (1.2-2.2) respectively. They need to mention this in text.in Abstract, Results and Discussion.

They note that associations of high strain jobs with lower odds ratio of continued work >63 years are significant in both unadjusted and fully adjusted models in blue-collar workers. In their report of high strain jobs being associated with increased odds of continuing to work in all and male white-collar workers, they say (line 281) this effect “persisted in fully adjusted models and in men.” Inspection of Table 4 shows that this (and ERI’s) association was not significant in analyses with Models 2 and 3 adjusted, which is noted in next sentence (lines 283-284). They need to correct line 281.

7. PLOS authors have the option to publish the peer review history of their article (what does this mean?). If published, this will include your full peer review and any attached files.

Reviewer #1: No

---

## [Author Response · Author response to Decision Letter 2]

19 Mar 2020

Reviewer #1 

Concern 1 & 2We deeply regret that we expressed ourselves unclear, but in the previously response letter we reported that the interaction effects between psychosocial variables and socioeconomic status (division: blue-/white-collar workers) were NOT significant anymore when entering the additional confounders requested in the previous revision by reviewer #2. So therefore, we prefer not to report these in the article. Also, if reporting on interactive effects for: psychosocial variables*other job variables; psychosocial variables * socioeconomic, then there is little justification to why we should not report interaction between main psychosocial variables and all covariates, and this would shift the focus of the study somewhat. As our study design focuses on psychosocial variables and relationships to continued work, we are not sure if this add much value, especially as the interactions mentioned are non-significant. Regarding adding the interaction because this is a justification for doing separate analyses of blue- and white-collar workers, we would argue that it is common practice to do separate analyses for these two groups. Moreover, it in the case of Volvo, it is clear that they have very different working environments. 

We do, nevertheless, appreciate the reviewer’s valuable suggestions regarding how to investigate the complexity in a retirement process as multiple variables have indeed a joint effect on an executed retirement or not. We are also very interested in such joint effects on psychosocial variables and socioeconomic markers, as similar topics have arisen in other of our research projects. We therefore plan for our next study to look at retirement in clusters of persons with specific set of several physical and psychosocial work conditions and socioeconomic markers (more than just blue-/white-collar worker, which may be too simplified). However, since the design used in the current paper is in accordance with general practice in analyses with job demand-control and effort reward imbalance (two of the authors main research focus is these psychosocial job variables) we felt it important to start with a first study using a standard design, and then build on those findings to make a more complex study in our next paper.

Regarding the comment on Table 4, that we neglected to report that high ERI is associated with extended work in all, and males - we did report this in the Results, but agree that the text was unclear. We have revised the text on page 14 to make it more clear. 

We have also added this information in Abstract – p 2, and in the Discussion – p 16. 

Regarding the mistake in line 281, we have corrected this mistake both in the Results – line 281, and in the Discussion - page 16. Once again, we are thankful for the thorough review.

---

## [Decision Letter · Decision Letter 3]

7 Apr 2020

PONE-D-19-15489R3

A sustainable working life in the car manufacturing industry: The role of psychosocial factors, gender and occupation

PLOS ONE

Dear Dr Gyllensten,

Thank you for submitting your manuscript to PLOS ONE. After careful consideration, we feel that it has merit but does not fully meet PLOS ONE’s publication criteria as it currently stands. Therefore, we invite you to submit a revised version of the manuscript that addresses the points raised during the review process.

After assessing this revised version of the manuscript, the Reviewer has expressed their satisfaction with most of the improvements and clarifications done in your last round of revisions. Nevertheless, an additional issue (that is important) has been raised and requires your attention. Please see the comments below.

Also, I noticed that the raw data supporting the study results is not already available in your submission. In this regard, please consider that PLOS Data policy is quite strict in this regard **("PLOS journals require authors to make all data necessary to replicate their study’s findings publicly available without restriction at the time of publication. When specific legal or ethical restrictions prohibit public sharing of a data set, authors must indicate how others may obtain access to the data")**, and it must be fully available in order to accept the paper for publication.

We would appreciate receiving your revised manuscript by May 22 2020 11:59PM. To enhance the reproducibility of your results, we recommend that if applicable you deposit your laboratory protocols in protocols.io, where a protocol can be assigned its own identifier (DOI) such that it can be cited independently in the future. For instructions see: http://journals.plos.org/plosone/s/submission-guidelines#loc-laboratory-protocols

We look forward to receiving your revised manuscript.

Kind regards,

Sergio A. Useche, Ph.D.

Academic Editor

PLOS ONE

Reviewers' comments:

Reviewer's Responses to Questions

**Comments to the Author**

1. If the authors have adequately addressed your comments raised in a previous round of review and you feel that this manuscript is now acceptable for publication, you may indicate that here to bypass the “Comments to the Author” section, enter your conflict of interest statement in the “Confidential to Editor” section, and submit your "Accept" recommendation.

Reviewer #1: (No Response)

2. Is the manuscript technically sound, and do the data support the conclusions?

Reviewer #1: Yes

3. Has the statistical analysis been performed appropriately and rigorously? 

Reviewer #1: Yes

4. Have the authors made all data underlying the findings in their manuscript fully available?

Reviewer #1: No

5. Is the manuscript presented in an intelligible fashion and written in standard English?

Reviewer #1: Yes

6. Review Comments to the Author

Reviewer #1: They do note in the Results and early in the Discussion that the finding of high strain (and ERI) association with more likelihood to continue work becomes nonsignificant in the fully adjusted models. This casts doubt on the validity of this association -- suggesting that the extensive discussion of it (lines 328-349, P. 17) should be omitted.

7. PLOS authors have the option to publish the peer review history of their article (what does this mean?). If published, this will include your full peer review and any attached files.

Reviewer #1: No

---

## [Author Response · Author response to Decision Letter 3]

9 Apr 2020

Comment from editor: I noticed that the raw data supporting the study results is not already available in your submission. In this regard, please consider that PLOS Data policy is quite strict in this regard ("PLOS journals require authors to make all data necessary to replicate their study’s findings publicly available without restriction at the time of publication. When specific legal or ethical restrictions prohibit public sharing of a data set, authors must indicate how others may obtain access to the data"), and it must be fully available in order to accept the paper for publication.

Response: In our response to reviewers submitted in October 2019, we included the following text. The statement includes information regarding how the data can be obtained. We hope that this is sufficient information regarding availability . 

“Data cannot be shared publicly because information about health and symptoms are regarded as sensitive information, and when sharing such data there has to be an approval from a Swedish Ethical committee (according to Swedish law). However, anonymised data is available with an approval from an ethical review board. For data requests, contact: Kristina Gyllensten, Department of occupational and environmental medicine, Gothenburg University, Box 414, 405 30, Gothenburg, Sweden. The name of the data set is ‘The Volvo Work Ability Study’. 

Reviewer #1: They do note in the Results and early in the Discussion that the finding of high strain (and ERI) association with more likelihood to continue work becomes nonsignificant in the fully adjusted models. This casts doubt on the validity of this association -- suggesting that the extensive discussion of it (lines 328-349, P. 17) should be omitted.

Response: Thank you for pointing this out. Lines 328-349 has now been deleted.

---

## [Editor Report · Decision Letter 4]

28 Apr 2020

A sustainable working life in the car manufacturing industry: The role of psychosocial factors, gender and occupation

PONE-D-19-15489R4

Dear Dr. Gyllensten,

We are pleased to inform you that your manuscript has been judged scientifically suitable for publication and will be formally accepted for publication once it complies with all outstanding technical requirements.

With kind regards,

Sergio A. Useche, Ph.D.

Academic Editor

PLOS ONE
---

## [Editor Report · Acceptance letter]

1 May 2020

PONE-D-19-15489R4 

A sustainable working life in the car manufacturing industry: The role of psychosocial factors, gender and occupation 

Dear Dr. Gyllensten:

I am pleased to inform you that your manuscript has been deemed suitable for publication in PLOS ONE. Congratulations! Your manuscript is now with our production department. 

With kind regards,

on behalf of

Dr. Sergio A. Useche 

Academic Editor

PLOS ONE